# Highly Specific Loop-Mediated Isothermal Amplification Using Graphene Oxide–Gold Nanoparticles Nanocomposite for Foot-and-Mouth Disease Virus Detection

**DOI:** 10.3390/nano12020264

**Published:** 2022-01-14

**Authors:** Jong-Won Kim, Kyoung-Woo Park, Myeongkun Kim, Kyung Kwan Lee, Chang-Soo Lee

**Affiliations:** 1Bionanotechnology Research Center, Korea Research Institute of Bioscience & Biotechnology (KRIBB), Daejeon 34141, Korea; zzagasj@gmail.com (J.-W.K.); qkrruddn94@kribb.re.kr (K.-W.P.); kmkun8510@naver.com (M.K.); lkk@kribb.re.kr (K.K.L.); 2Department of Biotechnology, University of Science & Technology (UST), Daejeon 34113, Korea; 3Department of Biomedical and Nanopharmaceutical Science, College of Pharmacy, Kyung Hee University, Seoul 02447, Korea

**Keywords:** loop-mediated isothermal amplification, foot-and-mouth disease virus, graphene oxide, gold nanoparticles, nanocomposites

## Abstract

Loop-mediated isothermal amplification (LAMP) is a molecular diagnosis technology with the advantages of rapid results, isothermal reaction conditions, and high sensitivity. However, this diagnostic system often produces false positive results due to a high rate of non-specific reactions caused by formation of hairpin structures, self-dimers, and mismatched hybridization. The non-specific signals can be due to primers used in the methods because the utilization of multiple LAMP primers increases the possibility of self-annealing of primers or mismatches between primers and templates. In this study, we report a nanomaterial-assisted LAMP method that uses a graphene oxide–gold nanoparticles (AuNPs@GO) nanocomposite to enable the detection of foot-and-mouth disease virus (FMDV) with high sensitivity and specificity. Foot-and-mouth disease (FMD) is a highly contagious and deadly disease in cloven-hoofed animals; hence, a rapid, sensitive, and specific detection method is necessary. The proposed approach exhibited high sensitivity and successful reduction of non-specific signals compared to the traditionally established LAMP assays. Additionally, a mechanism study revealed that these results arose from the adsorption of single-stranded DNA on AuNPs@GO nanocomposite. Thus, AuNPs@GO nanocomposite is demonstrated to be a promising additive in the LAMP system to achieve highly sensitive and specific detection of diverse diseases, including FMD.

## 1. Introduction

Foot-and-mouth disease virus (FMDV) is an RNA virus belonging to the genus *Aphthovirus* of the family *Picornaviridae* that causes a highly contagious and often fatal disease in cloven-hoofed animals, including cattle, sheep, goats, and swine [1]. The disease and restriction of international trade in animals to prevent the spread of epidemics to FMDV-free areas lead to huge economic losses in livestock industries [2]. FMDV has been immunologically classified as seven serotypes—O, A, C, Southern African Territories (SAT) 1, SAT 2, SAT 3, and Asia 1—with an antigenic spectrum of various strains within each serotype [3]. FMDV serotypes O and A have frequently broken out in East Asia, especially South Korea, since the early 2000s [4]. Thus, the early and accurate detection of FMDV serotypes O and A has been urgently needed.

For in vitro detection of FMDV, molecular biological methods, such as polymerase chain reaction (PCR) and loop-mediated isothermal amplification (LAMP), have been widely employed due to their high sensitivity, achieved through gene amplification of the target nucleic acids [5,6,7]. PCR has been considered the most powerful and standard molecular diagnostic tool [8,9,10,11]. However, this technique has several limitations due to the requirements of a time-consuming, laborious multiple-step process and thermal cycling equipment, resulting in a delay in diagnosis time [12,13]. In contrast, LAMP has been noted as a useful molecular diagnostic tool for the detection of diverse diseases due to its high sensitivity, specificity, rapidity, and simplicity [14]. Nevertheless, it suffers from non-specific amplifications and a resultant high rate of false positive reactions, which seriously restrict its point-of-care testing (POCT) application [15,16,17]. Recent advances have proved that nanomaterials such as gold nanoparticles (AuNPs) and graphene oxide (GO) can effectively inhibit undesired amplification [16,18,19]. AuNPs and GO have been proposed as biomedical nanomaterials for diagnosis, treatment, and drug delivery, as well as chemical nanomaterial for catalysts and sensors, due to unique chemical and physical features, including the ability to reduce non-specific signals effectively [20,21,22,23,24,25,26,27,28,29].

In this study, we developed a nanomaterial-assisted LAMP system for highly sensitive and specific detection of FMDV using a GO–AuNPs (AuNPs@GO) nanocomposite. In order to demonstrate the effect of AuNPs@GO as a LAMP additive, FMDV serotypes O and A were chosen as target viruses; we found that AuNPs@GO nanocomposite can successfully inhibit false amplification caused by wrong hybridization. Additionally, the sensitivity of our LAMP system using AuNPs@GO was increased by ~100-fold for FMDV serotype A genes, and its speed was faster by 7 min for FMDV serotype O genes, compared to that using GO. To verify how AuNPs@GO nanocomposite works in the LAMP system, we performed a mechanism study through analysis of the interaction between LAMP reagents, including primers, *Bst* DNA polymerase, GO, and AuNPs@GO nanocomposite. The strong interaction between AuNPs@GO and primers was precisely identified, verifying that this remarkable technique would be useful in practice for rapid and sensitive virus detection in early stages of various infectious diseases, including FMDV.

## 2. Materials and Methods

### 2.1. Materials

Tetrachloroauric (III) acid trihydrate (HAuCl_4_ 3H_2_O, Sigma-Aldrich, St. Louis, MO, USA), trisodium citrate (Na_3_C_6_H_5_O_7_, Sigma-Aldrich, St. Louis, MO, USA), and GO (Graphene Supermarket, Calverton, NY, USA) were used as purchased. *Bst* DNA polymerase, 10X of ThermoPol buffer, 100 mM magnesium sulfate (MgSO_4_), and 10 mM dNTP were obtained from New England BioLabs (Ipswich, MA, USA). SYBR green I fluorescent nucleic acid was purchased from LONZA (Rockland, ME, USA). Ultrapure water was used throughout this study.

### 2.2. Preparation of AuNPs@GO Nanocomposite

The AuNPs@GO nanocomposite with an elemental composition ratio (carbon: gold) of 0.84: 11.7 (wt %) was prepared through in situ growth of AuNPs on GO sheets [30,31,32]. Briefly, 5 mL of GO solution (1.5 mg/mL) was prepared by sonication for 5 min, then 100 mL HAuCl_4_·3H_2_O (0.1 mg/mL) was added and the mixture was stirred for 30 min at 400 rpm. This procedure allows gold ions to adsorb onto the GO sheets. Next, the mixture was heated to 90 °C, followed by adding 1 mL trisodium citrate (0.035 g/mL) as a reducing agent. When the temperature was above 90 °C, the mixture was not well-dispersed. After heating for 10 min, the mixture was stirred for 1 h to complete the reaction. The prepared solution was cooled at ambient temperature, presenting a deep red color. Finally, the resultant solution was purified three times by centrifugation at 3940× g (15 min) with distilled water to remove free AuNPs.

### 2.3. Characterization of AuNPs@GO Nanocomposite

The characterization of the prepared AuNPs@GO nanocomposite was conducted using UV-vis spectroscopy, field-emission transmission electron microscopy (FE-TEM), and zeta potential. The optical spectra of AuNPs@GO nanocomposite were collected using a UV-vis spectrophotometer (Optizen pop, Mecasys, Daejeon, South Korea). Morphology and zeta potentials were determined using a Talos F200X FE-TEM (FEI Co., Hillsboro, OR, USA) at 200 kV and dynamic light scattering (DLS; Zetasizer Nano ZS, Malvern Instruments, Malvern, UK), respectively.

### 2.4. LAMP Assays

The protocol of LAMP assays was followed using a basic LAMP working system (New England BioLabs) at 65 °C for 60 min. The LAMP mixture (total volume 25 µL) consisted of the following reagents: 1X ThermoPol buffer for *Bst* DNA polymerase, large fragment, MgSO_4_ (total 8 mM), and dNTP mix (1.4 mM each); FIP/BIP primers (1.6 µM), F3/B3 primers (0.2 µM), LoopF/BoopF (0.4 µM), and *Bst* DNA polymerase, large fragment (320 U/mL); graphene oxide (5 µL), AuNPs (5 µL), or AuNPs@GO nanocomposite (5 µL) solutions; 1 µL target or ultrapure water (no target); 0.4X SYBR green I; and ultra-pure water to a final volume of 25 µL. FMDV genes used in this study were prepared according to a previous report [33]. Briefly, sequences of FMDV serotype O and A genes were selected from part of the RNA-dependent RNA polymerase sequences which were originated from O/Andong/KOR/2010 and A/Pocheon/001/KOR/2010 strains, respectively. The serotype O and A genes were cloned into pET-21a plasmids (Novagen, Inc., Madison, WI, USA) and the resulting plasmids were transformed into DH5α *E. coli* cells (RBC Bioscience Corp., New Taipei City, Taiwan). The transformed cells were grown in a LB medium containing ampicillin (100 μg/mL) at 37 °C overnight. Plasmids were prepared from cultured and harvested cells using extraction kits (Bioneer, Daejeon, Korea). All LAMP primers (forward F3, FIP, and LF and reverse B3, BIP, and LB) were designed using PrimerExplorer V5, primer design software that specializes in LAMP (http://primerexplorer.jp/lampv5e/index.html (accessed on 10 January 2022), Eiken Chemicals Corporation, Tokyo, Japan). The prepared primer sets of FMDV are shown in Appendix A.

## 3. Results and Discussion

### 3.1. Characterization of AuNPs@GO Nanocomposite

The AuNPs@GO nanocomposite was synthesized by in situ reduction through Au^3+^ ion seed-mediated growth on GO sheets, and its successful preparation was demonstrated using UV-vis spectroscopy, zeta potentials, and TEM (Figure 1). The UV-vis spectra of AuNPs, GO, and AuNPs@GO nanocomposite are shown in Figure 1a. The prepared AuNPs@GO nanocomposite had two absorption peaks regarded as the localized surface plasmon resonance (LSPR) spectrum at 520 nm and 225 nm. The absorption peak at 520 nm indicates a typical LSPR peak originating from AuNPs (ca. 15 nm in diameter) grown on GO sheets. The spectrum of GO at 225 nm represents π–π* transitions for the C=C bonds and n–π* transitions for the C=O bonds of GO [34,35]. Thus, the successful adsorption of AuNPs onto GO sheets was confirmed by observation of optical characteristics of both AuNPs and GO in the AuNPs@GO nanocomposite.

Further, the zeta potentials of GO, AuNPs, and AuNPs@GO nanocomposite were measured to evaluate the changes of surface charge in the process of AuNPs@GO nanocomposite preparation. As shown in Figure 1b, the zeta potentials of GO, AuNPs, and AuNPs@GO nanocomposite were estimated to be −44.37 ± 0.77, −15.43 ± 2.73, and −36.40 ± 0.64 mV, respectively. These findings in the zeta potential demonstrated the successful formation of AuNPs with weaker negative charge on the relatively strongly negatively charged GO sheets.

To further understand the morphological properties of the AuNPs@GO nanocomposite, FE-TEM measurements were conducted. As shown in Figure 1c, the spherical AuNPs were randomly distributed and localized on the surface of the GO sheets with a diameter of 15.6 ± 1.3 nm, and morphological deformation was not observed.

Based on these results, we found that GO could play a role as a supporting substrate for the nanocomposite, which had characteristics of both GO and AuNPs. The findings also confirmed that GO could prevent self-aggregation of AuNPs@GO nanocomposite by acting as a building block, while allowing excellent water dispersibility.

### 3.2. LAMP Assays Using AuNPs@GO Nanocomposite

The AuNPs@GO nanocomposite was used as a component of the LAMP reaction. In the traditional LAMP reaction, the use of multiple primers—typically six kinds—occasionally has caused the generation of non-specific signals through formation of primer dimers, hairpin structures, or mismatches between primers and templates, all of which can produce incorrect results [36]. Therefore, reducing false positives is crucial for precise diagnosis. 

We investigated the effect of AuNPs@GO nanocomposite during the detection of FMDV genes using the LAMP reaction. To confirm the performance of AuNPs@GO nanocomposite, we compared the reduction of non-specific signals observed when GO and AuNPs@GO nanocomposite were used. The use of citrate AuNPs as a control was excluded in this experiment due to formation of aggregates in the LAMP reaction buffer (data not shown). We designed the primers for FMDV serotype O and A genes and conducted LAMP assays by applying various amounts of GO and AuNPs@GO nanocomposite from 0 to 70 ng.

As shown in Figure 2, non-specific signals for blanks (for both FMDV serotype O and A assays), which might be generated by primers, were gradually delayed when 10 ng of GO or AuNPs@GO nanocomposite was added, and completely disappeared when 20 ng or more was added.

In contrast, when more than 20 ng of GO or AuNPs@GO nanocomposite was added for FMDV target genes, the positive signals were appreciably delayed as well. Notably, we found that the appropriate amount of GO and AuNPs@GO nanocomposite had a significant effect on improvement of the LAMP reaction. Further, comparing the LAMP efficiency of GO and AuNPs@GO nanocomposite for FMDV genes, the use of AuNPs@GO nanocomposite generated the positive signals approximately 9 and 5 min faster than that of GO for FMDV serotype O and A genes, respectively (Figure 2, bottom panels).

Typically, GO and AuNPs can interact with primers through π–π and electrostatic interactions, respectively [37]. AuNPs@GO nanocomposite have a larger surface area than GO or AuNPs due to immobilization of AuNPs onto GO sheets. Additionally, it was previously reported that AuNPs@GO nanocomposite could bind to primers through a stronger affinity compared to GO or AuNPs [38]. These indicate that AuNPs@GO nanocomposite can play a crucial role as a stabilizer and enhancer in the LAMP reaction. As a result, 20 ng was determined to be the optimal content of GO and AuNPs@GO nanocomposite for the LAMP reaction; thus, we applied this number of nanomaterials to subsequent LAMP experiments.

Next, we evaluated the detection sensitivity for FMDV genes. Figure 3 presents the variation of fluorescent signals with varying concentrations of DNA template. Upon determining sensitivity, the threshold was set to be 1000 RFU (relative fluorescence units), which was above the background signal. First, for confirmation of the effect of GO and AuNPs@GO nanocomposite, LAMP reactions were performed without additives. The detection limit without additives could not be determined due to non-specific amplification and no tendency for fluorescence to change with the gene concentration (Appendix A). The detection limits of FMDV serotype O and A genes with addition of AuNPs@GO nanocomposite were estimated as 1 fg and 10 fg, respectively, whereas those with addition of GO were 1 fg and 1 pg, respectively. Compared to GO, the LAMP assay for FMDV serotype A genes using AuNPs@GO nanocomposite showed a 100-fold improvement in sensitivity.

To compare the detection efficiency for FMDV genes of GO and AuNPs@GO nanocomposite, time-dependent standard curves were obtained using the cycle quantitation values (Cq) of LAMP assays. As shown in Figure 4, the use of GO and AuNPs@GO nanocomposite produced linear relationships from 1 fg to 1 ng using FMDV serotype O genes. Additionally, the results of GO and AuNPs@GO with FMDV serotype A ranged from 1 pg to 1 ng and from 10 fg to 1 ng, respectively. Further, the fluorescent signals for AuNPs@GO nanocomposite with FMDV serotype O were assessed to be approximately 7 min faster than those with GO at all target concentrations.

These findings suggest that both GO and AuNPs@GO nanocomposite served as efficient components for reduction of non-specific amplifications, but AuNPs@GO nanocomposite have less effect on the LAMP reaction with templates than GO. It might be considered that AuNPs@GO nanocomposite has strong affinity for certain components composing the LAMP mixture, especially primers, reducing the probability of non-specific reactions. These results demonstrate that AuNPs@GO nanocomposite could be useful as a nanomaterial additive for sensitive detection of FMDV genes as well as reduction of non-specific amplification in the LAMP system.

### 3.3. Specificity for Serotype O and Serotype A

FMDV has been classified into seven serotypes, of which serotypes O and A have recently spread around the world, especially in South Korea. Therefore, we designed a strategy to obtain high specificity through the cross-reaction between serotypes. The experiments were conducted using Lambda DNA and *E. coli* genomic DNA as controls to determine the effect of AuNPs@GO nanocomposite on the specificity.

As shown in Figure 5a,b, the addition of AuNPs@GO nanocomposite specifically generated fluorescence signals for the target DNA (serotypes O and A) at 28 and 29 min, respectively, displaying no signal for Lambda DNA and *E. coli* DNA. Meanwhile, we found that without AuNPs@GO nanocomposite, the target DNA produced fluorescence signals at 24 and 25 min, and Lambda DNA and *E. coli* DNA produced signals at 22 and 24 min, respectively. These results indicate that AuNPs@GO nanocomposite specifically yielded positive detection with serotype O and A primers, not allowing amplification of other DNA templates. These significant differences demonstrate that our proposed method could identify different types of FMDV by specifically detecting serotypes O and A DNA, showing an excellent specificity for target DNA.

### 3.4. Mechanism of AuNPs@GO Nanocomposite for LAMP Assays

Recently, various nanomaterials including GO and AuNPs have been actively studied as an enhancer in the PCR system. GO and AuNPs have been known to interact with components such as primers and DNA polymerase in the PCR mixture through π–π stacking and electrostatic interaction, resulting in elimination of non-specific amplification and enhancement of detection performance. 

To study how AuNPs@GO nanocomposite works in the LAMP system, we investigated the binding mode of AuNPs@GO nanocomposite with primers and *Bst* DNA polymerase using UV-vis spectroscopy. In UV-vis spectra of GO and AuNPs@GO nanocomposite, a strong absorption peak observed at 225 nm arises from the π–π* transition of the C=C bond (Figure 6a). Upon mixing *Bst* DNA polymerase with each solution containing GO and AuNPs@GO nanocomposite, an absorption peak at 280 nm was observed, which represents absorption peak of a protein (Figure 6b,c). This verifies that GO and AuNPs@GO nanocomposite have an affinity for *Bst* DNA polymerase. Additionally, when LAMP primers were included in each solution, the appearance of the absorption peak at 260 nm presents the adsorption of primers on the surface of GO and AuNPs@GO nanocomposite, in which a difference in increase of the intensity of the absorption peaks was clearly observed. Whereas AuNPs@GO nanocomposite increased by 2.11 times the absorption peak around 260 nm, GO showed a slight increase of 1.32 times after addition of primers (Figure 6b,c). 

These results demonstrate that AuNPs@GO nanocomposite has a stronger interaction with primers than GO. Additionally, as shown in Figure 6d, the representative wavelength peak of AuNPs on GO sheets at 523 nm were slightly shifted to 525 nm with addition of *Bst* DNA polymerase, which indicates an interaction between AuNPs and *Bst* DNA polymerase. 

In contrast, when both primers and *Bst* DNA polymerase were added, the absorption peak was similar to that with AuNPs@GO nanocomposite and primers. This indicates that AuNPs are susceptible to binding to primers rather than *Bst* DNA polymerase. As a result, it revealed that AuNPs@GO nanocomposite have stronger binding affinity to primers than GO, as well as preferring interaction with primers rather than *Bst* DNA polymerase, as presented in a previous report [38]. 

Our results suggest that AuNPs@GO nanocomposite has a larger surface area than GO because AuNPs are incorporated onto GO sheets, inducing an enhancement in interaction with the LAMP mixture, especially primers, through synergistic effects of GO and AuNPs. Taken together, these findings demonstrate that AuNPs@GO could provide a valuable feasibility for efficient performance of a LAMP assay system.

The LAMP platform has been spotlighted as an alternative to PCR due to its fast, convenient, and highly specific molecular diagnostic ability for certain diseases [39,40,41]. Additionally, POCT using the LAMP platform can provide rapid results, thus supporting prevention of the spread of highly infectious diseases. However, the use of the LAMP platform has sometimes been restricted due to frequent generation of non-specific signals, resulting in false positive results and complexity in designing primers. 

We aimed to address these issues by using AuNPs@GO nanocomposite in the LAMP reaction to reliably detect FMDV. As depicted in Figure 7, the reduction of non-specific signals is realized by AuNPs@GO nanocomposite inhibiting undesirable amplifications. Non-specific signals in the traditional LAMP system have been known to be associated with high concentration of multiple primers, which often form hairpin structures, self-dimers, and mismatched hybridizations.

One solution is to reduce the primer concentration; however, this could reduce sensitivity and increase detection time. Therefore, the use of nanomaterials as components in the LAMP mixture could be considered as a simple and convenient approach. 

For instance, it was reported that AuNPs could enhance the specificity in LAMP assays by inserting a pre-incubation step before initiation of the LAMP reaction, which is called the hot-start LAMP technique [19]. Citrate AuNPs are known to exhibit high affinity toward nitrogen atoms in the four bases and weakly negatively charged backbones of ssDNA rather than dsDNA via electrostatic effects.

In addition, GO has various functional groups, such as hydroxyl, carboxyl, and epoxy groups, that can interact with primers and *Bst* DNA polymerase through π–π stacking and electrostatic interactions [22]. In this regard, we developed an AuNPs@GO nanocomposite, creating a synergetic effect by combining GO and AuNPs. In addition to the interaction mechanism between AuNPs@GO nanocomposite and components in the LAMP mixture, the large surface area of AuNPs@GO nanocomposite contributes to reduction of non-specific amplification, as well as enhanced efficiency of the LAMP assay. By conducting a mechanism study, we demonstrated that AuNPs@GO nanocomposite can strongly suppress undesirable amplifications. In summary, first, AuNPs@GO can act as a building block, which means that the stability of LAMP components including primers, DNAs and polymerase can be improved through interaction with AuNPs@GO during LAMP reactions. Additionally, the strong interaction between AuNPs@GO and primers may prevent mismatched hybridization of primers before pairing with target DNAs. Finally, the weaker negative charge of AuNPs@GO may facilitate access of target DNAs or primers more than GO, resulting in high sensitivity and specificity. Therefore, AuNPs@GO nanocomposite can be useful for precise and efficient FMDV detection using the LAMP method.

## 4. Conclusions

We developed a LAMP assay for detection of FMDV with reduced non-specific amplification using AuNPs@GO nanocomposite. AuNPs@GO nanocomposite was successfully synthesized and the optimal concentration for the LAMP assay was determined. Compared with GO, AuNPs@GO nanocomposite exhibited inhibition of non-specific amplification, 7 min faster reaction for FMDV serotype O, and 100-fold higher sensitivity for FMDV serotype A, enabling precise detection of FMDV genes. We also demonstrated the binding mode of AuNPs@GO nanocomposite with the LAMP mixture; the AuNPs@GO nanocomposite has a strong affinity for primers. Therefore, our system based on AuNPs@GO nanocomposite could provide the opportunity for detection of FMDV using other gene amplification tools.

## Figures and Tables

**Figure 1 nanomaterials-12-00264-f001:**
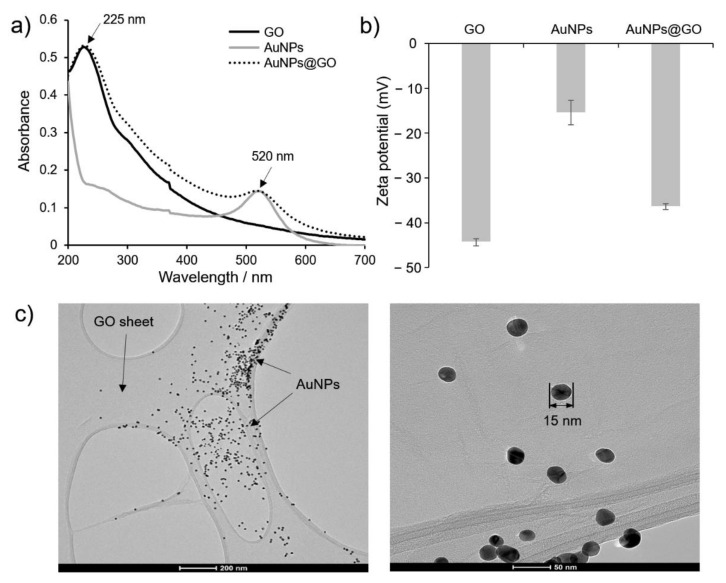
Characterization of the prepared AuNPs@GO nanocomposite. (**a**) UV-vis spectra and (**b**) zeta potentials of graphene oxide (GO), gold nanoparticles (AuNPs), and GO–AuNPs (AuNPs@GO) nanocomposite, and (**c**) transmission electron microscopy images of AuNPs@GO nanocomposite.

**Figure 2 nanomaterials-12-00264-f002:**
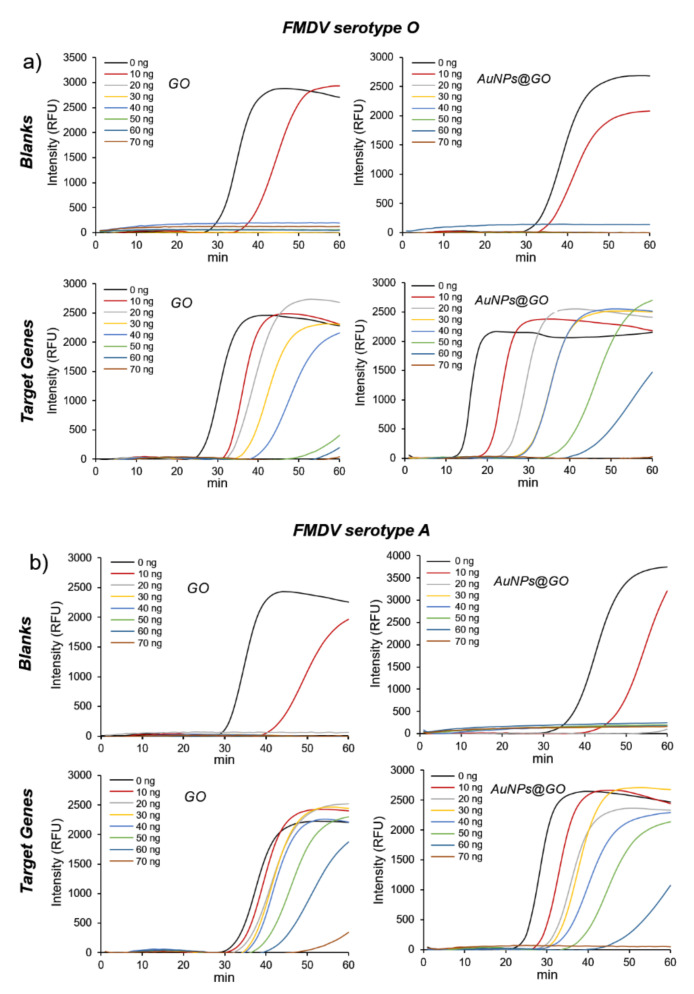
Effect of graphene oxide (GO) and GO–AuNPs (AuNPs@GO) nanocomposite on loop-mediated isothermal amplification (LAMP) reaction using foot-and-mouth disease virus (FMDV) genes. Real-time fluorescence measurement of LAMP reaction for blanks (top) and target genes (bottom) using (**a**) FMDV serotype O and (**b**) FMDV serotype A genes. Left and right panels, GO and AuNPs@GO nanocomposite, respectively.

**Figure 3 nanomaterials-12-00264-f003:**
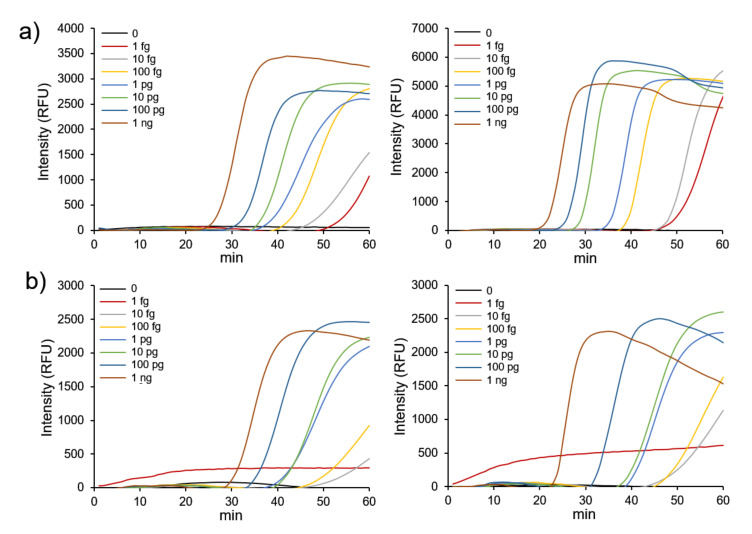
Effect of foot-and-mouth disease virus (FMDV) gene concentration on the loop-mediated isothermal amplification (LAMP) reaction using FMDV genes. Real-time fluorescence monitoring of the LAMP reaction using (**a**) FMDV serotype O and (**b**) FMDV serotype A genes. Left and right panels, graphene oxide (GO), and GO–AuNPs (AuNPs@GO) nanocomposite, respectively.

**Figure 4 nanomaterials-12-00264-f004:**
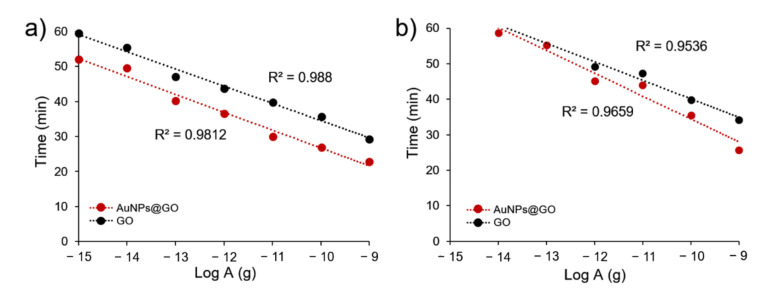
Standard curves of detection time versus amount of gene material using (**a**) FMDV serotype O and (**b**) FMDV serotype A genes.

**Figure 5 nanomaterials-12-00264-f005:**
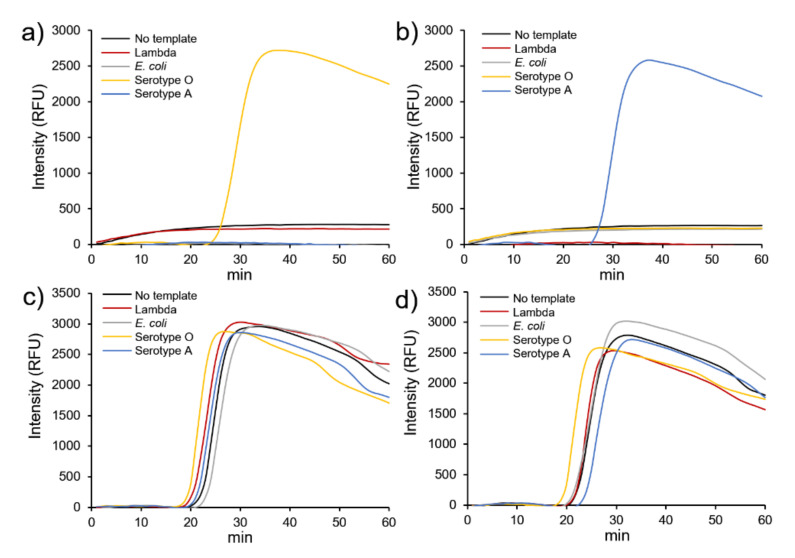
Specificity of loop-mediated isothermal amplification (LAMP) with GO–AuNPs (AuNPs@GO) nanocomposite using serotype O- and A-specific primers and 100 pg foot-and-mouth disease virus (FMDV) genes. (**a**) Serotype O-specific primers and (**b**) serotype A-specific primers with AuNPs@GO nanocomposite, (**c**) serotype O-specific primers, and (**d**) serotype A-specific primers without AuNPs@GO nanocomposite.

**Figure 6 nanomaterials-12-00264-f006:**
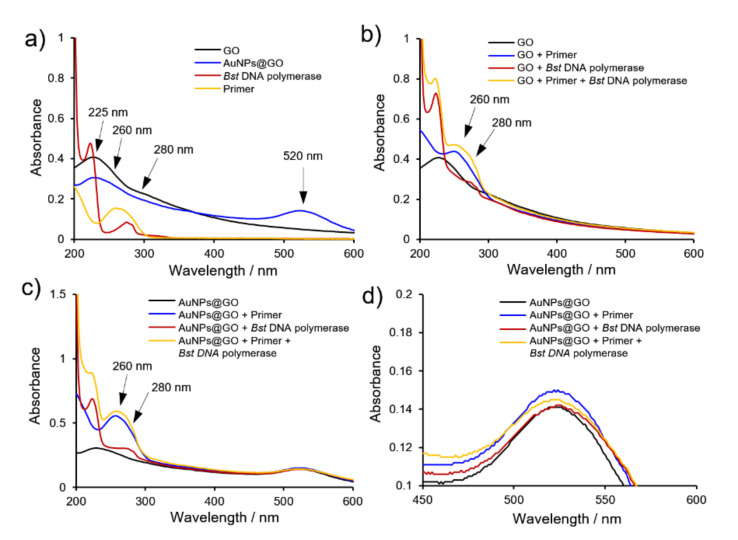
(**a**) UV-vis spectra of graphene oxide (GO), GO–AuNPs (AuNPs@GO) nanocomposite, *Bst* DNA polymerase, and LAMP primers. (**b**,**c**) Absorbance changes on addition of primer, *Bst* DNA polymerase, and both in b) GO and (**c**) AuNPs@GO nanocomposite. (**d**) Enlargement of absorption peak of AuNPs on GO sheets from (**c**).

**Figure 7 nanomaterials-12-00264-f007:**
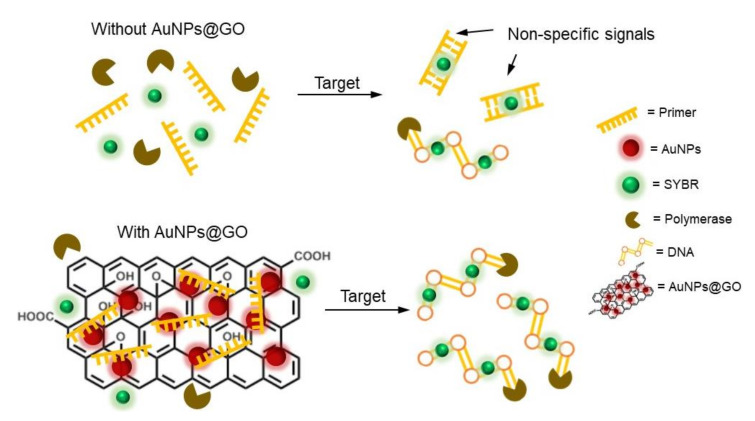
Schematic illustration of non-specific signal reduction using GO–AuNPs (AuNPs@GO) nanocomposite in a loop-mediated isothermal amplification (LAMP) system.

## Data Availability

All other relevant data are available from the corresponding author upon reasonable request.

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
