# Peer review of "Highly Specific Loop-Mediated Isothermal Amplification Using Graphene Oxide–Gold Nanoparticles Nanocomposite for Foot-and-Mouth Disease Virus Detection"

_nanomaterials, 2022, doi:10.3390/nano12020264_

Round 1
Reviewer 1 Report
In this paper, the authors report a nanomaterial assisted lamp method, which uses graphene oxide gold nanoparticle nanocomposites to detect foot-and-mouth disease virus with high sensitivity and specificity. Compared with the traditional lamp analysis method, this method has higher sensitivity and successfully reduces nonspecific signals. In addition, a mechanism study shows that these results come from the adsorption of single stranded DNA on molecular sieve AuNPs@GO Nanocomposites. AuNPs@GO nanocomposite is demonstrated to be a promising additive in the LAMP system to achieve highly sensitive and specific detection of diverse diseases including FMD. There are some issues which the authors should address them before acceptance process of the paper. Here are my comments:
- What are the advantages of this job over other jobs? The author is advised to make a table for comparison.
- The size of gold nanoparticles should be very important for the detection effect. The author needs to count the size of gold nanoparticles.
- The authors of some related articles need to mention the application of graphene oxide and related nanostructures, such as:
DOI: 10.1016/j.electacta.2019.135196; DOI: 10.1016/j.snb.2018.01.068;
DOI: 10.1039/c7ra02198d; DOI: 10.1021/acssuschemeng.5b00383
Author Response
Replies to the reviewer 1’s comments
We are thankful for valuable comments of the reviewers, which helped us to correct errors and strengthen up our manuscript.
Reviewer #1 : In this paper, the authors report a nanomaterial assisted lamp method, which uses graphene oxide gold nanoparticle nanocomposites to detect foot-and-mouth disease virus with high sensitivity and specificity. Compared with the traditional lamp analysis method, this method has higher sensitivity and successfully reduces nonspecific signals. In addition, a mechanism study shows that these results come from the adsorption of single stranded DNA on molecular sieve AuNPs@GO Nanocomposites. AuNPs@GO nanocomposite is demonstrated to be a promising additive in the LAMP system to achieve highly sensitive and specific detection of diverse diseases including FMD. There are some issues which the authors should address them before acceptance process of the paper. Here are my comments:
- What are the advantages of this job over other jobs? The author is advised to make a table for comparison.
Response : In this study, we demonstrated that a nanomaterial-assisted LAMP system using AuNPs@GO nanocomposite clearly represented high sensitivity and specificity compared with conventional LAMP system by reducing non-specific amplification, which gives rise to false positive signals or misdiagnosis.
As mentioned in the manuscript, nanomaterials, such as AuNPs, GO, and pullulan, have been often introduced to overcome some drawbacks in conventional LAMP system. Thus, we proposed the use of AuNPs@GO nanocomposite for the combination of advantageous properties of AuNPs and GO, followed by showing the best performance compared with single materials, AuNPs and GO in our experimental conditions. This resulted in that AuNPs@GO nanocomposite showed a superior performance in sensitivity and specificity of FMDV detection. Unfortunately, it is hard to provide the table for comparison of advantages with other studies such as performance.
- The size of gold nanoparticles should be very important for the detection effect. The author needs to count the size of gold nanoparticles.
Response : The size of AuNPs on GO sheets was mentioned on Figure 1 in the manuscript. Briefly, the size of AuNPs on the surface of GO sheets was estimated as the average of AuNPs on GO sheets in TEM images. Additionally, we have applied the different size of AuNPs (15, 25 and 40 nm) and finally found an outstandinfg amplification performance when 15 nm AuNPs were used.
- The authors of some related articles need to mention the application of graphene oxide and related nanostructures, such as:
DOI: 10.1016/j.electacta.2019.135196; DOI: 10.1016/j.snb.2018.01.068;
DOI: 10.1039/c7ra02198d; DOI: 10.1021/acssuschemeng.5b00383
Response : As the reviewer commented, we included references (reference # 26-29) in the revised manuscript (page 2, line 56-59).
“ AuNPs and GO have been proposed as biomedical nanomaterials for diagnosis, treatment, and drug delivery, as well as chemical nanomaterial for catalysts and sensors, due to unique chemical and physical features, including the ability to reduce non-specific signals effectively [20-29].

Reviewer 2 Report
In this manuscript, the authors used graphene oxide (GO) and graphene oxide–gold nanoparticles (AuNPs@GO) to improve the performance of loop-mediated isothermal amplification (LAMP) for foot-and-mouth disease virus (FMDV) detection. Basically, this study is generally simple and quite interesting. However, numerous issues must be resolved before the consideration of publication.
- In line 52-54, the authors have introduced the application of AuNPs and GO for biomedical applications such as diagnosis-related applications. However, the descriptions of their advantages for these applications are not specific enough. Also, what not choosing other 2D nanomaterials such as MoS2 (e.g., J. Comp. Sci. 2021, 5(7):190) for the viral amplification diagnosis?
- In line 54, what kind of undesired amplification? Also, in line 56, how these features reduce non-specific signals effectively?
- In line 58-61, please provide the rationale for choosing AuNP@GO.
- The results show the characterization of AuNPs alone in the result section but the methodology only mention the synthesis of AuNP@GO nanocomposite
- In line 136-138, page 3, the zeta potentials of GO and Au NPs were -44 and -15 mV. However, the authors claimed the AuNPs@GO (-36.4 mV) being more negative that was not true.
- In line 172, page 4, there was no data of 20 mg shown in the figures. Should it be 20 ng?
- In Figure 6, the word polymerase should be added to the “Bst DNA” labelling to avoid confusion.
- “In contrast, when both primers and Bst DNA polymerase were added, the absorption peak almost corresponded to that with AuNPs@GO nanocomposite and primers.” However, the peak at 280 nm was covered by 260 nm.
- In line 281-283, page 11, since the Au NPs were also negative, please check if it is proper to claim that there is electrostatic interaction with negative DNA.
- “Non-specific signals in the traditional LAMP system has been known to be associated with high concentration of multiple primers, which often form hairpin structures, self-dimers, and mismatched hybridizations” A simplified version of this sentence can be added to the abstract for indicating its specific mechanism.
- The image resolution of this manuscript should be improved. The English language of this manuscript can be further polished.
Author Response
Replies to the reviewer 2’s comments
We are thankful for valuable comments of the reviewers, which helped us to correct errors and strengthen up our manuscript.
Reviewer #2 : In this manuscript, the authors used graphene oxide (GO) and graphene oxide–gold nanoparticles (AuNPs@GO) to improve the performance of loop-mediated isothermal amplification (LAMP) for foot-and-mouth disease virus (FMDV) detection. Basically, this study is generally simple and quite interesting. However, numerous issues must be resolved before the consideration of publication.
- In line 52-54, the authors have introduced the application of AuNPs and GO for biomedical applications such as diagnosis-related applications. However, the descriptions of their advantages for these applications are not specific enough. Also, what not choosing other 2D nanomaterials such as MoS2 (e.g., J. Comp. Sci. 2021, 5(7):190) for the viral amplification diagnosis?
Response : The detailed advantages for these applications were mentioned in results and discussion in the manuscript (page 9, line 234-238).
“Recently, various nanomaterials including GO and AuNPs have been actively studied as an enhancer in the PCR system. GO and AuNPs have been known to interact with components such as primers and DNA polymerase in the PCR mixture through π–π stacking and electrostatic interaction, resulting in elimination of non-specific amplification and enhancement of detection performance.”
In order to prevent overlapping contents, it is provided as a reference in the previous part and mentioned in detail in the discussion. The 2D nanomaterials such as MoS2 was already indicated in reference # 18. Briefly, MoS2 prevents non-specific amplification well, but also hinders Bst polymerase acivity, which hinders target amplification. For this reason, MoS2 has proven to be less capable than GO. Instead of using MoS2, GO-AuNP was used to develop something better than GO.
- In line 54, what kind of undesired amplification? Also, in line 56, how these features reduce non-specific signals effectively?
Response : As mentioned in the manuscript, undesired amplification means non-specific amplification (page 2, line 52-54).
“Nevertheless, it suffers from non-specific amplifications and a resultant high rate of false positive reactions, which seriously restrict its point-of-care testing (POCT) application [15-17].”
The feature of reducing non-specific signal was indicated in results and discussion in the manuscript (page 5, line 190-195).
“Typically, GO and AuNPs can interact with primers through π–π and electrostatic interactions, respectively [37]. AuNPs@GO nanocomposite have a larger surface area than GO or AuNPs due to immobilization of AuNPs onto GO sheets. Also, it was previously reported that AuNPs@GO nanocomposite could bind to primers through a stronger affin-ity compared to GO or AuNPs [38]. These indicate that AuNPs@GO nanocomposite can play a crucial role as a stabilizer and enhancer in the LAMP reaction.”
- In line 58-61, please provide the rationale for choosing AuNP@GO.
Response : The rationale for choosing AuNP@GO was indicated in results and discussion in the manuscript (page 5, line 190-195).
“Typically, GO and AuNPs can interact with primers through π–π and electrostatic interactions, respectively [37]. AuNPs@GO nanocomposite have a larger surface area than GO or AuNPs due to immobilization of AuNPs onto GO sheets. Also, it was previously reported that AuNPs@GO nanocomposite could bind to primers through a stronger affin-ity compared to GO or AuNPs [38]. These indicate that AuNPs@GO nanocomposite can play a crucial role as a stabilizer and enhancer in the LAMP reaction.”
Recent advances have proved that nanomaterials such as gold nanoparticles (AuNPs) and graphene oxide (GO) can effectively inhibit undesired amplification. For this reason, the hybrid composites have been created to use the advantage of these nanometerials synergically.
- The results show the characterization of AuNPs alone in the result section but the methodology only mention the synthesis of AuNP@GO nanocomposite.
Response : The characterization of AuNPs in the result section describes the properties of AuNPs on the surface of GO contained in AuNP@GO nanocomposite.
- In line 136-138, page 3, the zeta potentials of GO and Au NPs were -44 and -15 mV. However, the authors claimed the AuNPs@GO (-36.4 mV) being more negative that was not true.
Response : As the reviewer commented, we corrected the wrong sentences. (page 3, line 148-150)
“These findings in the zeta potential demonstrated the successful formation of AuNPs with weaker negative charge on the relatively strongly negatively charged GO sheets.”
- In line 172, page 4, there was no data of 20 mg shown in the figures. Should it be 20 ng?
Response : As the reviewer commented, we corrected the wrong sentences. (page 5, line 183)
“20 mg → 20 ng”
- In Figure 6, the word polymerase should be added to the “Bst DNA” labelling to avoid confusion.
Response : As the reviewer commented, we corrected the wrong sentence.(page 10, Figure 6)
“Bst DNA → Bst DNA polymerase”
- “In contrast, when both primers and Bst DNA polymerase were added, the absorption peak almost corresponded to that with AuNPs@GO nanocomposite and primers.” However, the peak at 280 nm was covered by 260 nm.
Response : We apologize for the confusion. We modified “almost corresponded to that with” to “similar to that with”. (page 10, line 258-259)
“In contrast, when both primers and Bst DNA polymerase were added, the absorption peak similar to that with AuNPs@GO nanocomposite and primers.”
- In line 281-283, page 11, since the AuNPs were also negative, please check if it is proper to claim that there is electrostatic interaction with negative DNA.
Response : We apologize for the confusion. We wanted to describe that ssDNA is less negatively charged than dsDNA, so electrostatic effect is less. As the reviewer commented, we corrected the wrong sentences. (page 11, line 292-294)
“Citrate AuNPs are known to exhibit high affinity toward nitrogen atoms in the four bases and weakly negatively charged backbones of ssDNA rather than dsDNA via electrostatic effects.”
- “Non-specific signals in the traditional LAMP system has been known to be associated with high concentration of multiple primers, which often form hairpin structures, self-dimers, and mismatched hybridizations” A simplified version of this sentence can be added to the abstract for indicating its specific mechanism.
Response : As the reviewer commented, we added description regarding abstract. (page 1, line 15-17)
“However, this diagnostic system often produces false positive results due to a high rate of non-specific reactions caused by formation of hairpin structures, self-dimers, and mismatched hybridization.”
- The image resolution of this manuscript should be improved. The English language of this manuscript can be further polished.
Response : As the reviewer suggested, we improved the image resolution and the English language of this manuscript was revised.

Reviewer 3 Report
Authors describe AuNPs@GO nanocomposite assisted LAMP assay for detection of FMDV as a mode to increase sensitivity and specificity of the viral genome detection. The current work is a continuation of the authors' previous studies on using AuNPs@GO as a PCR enhancer. The current work can be a useful contribution to improvement the LAMP, which still needs optimization. I recommend publishing this work with minor corrections.
The FMDV is a RNA virus. Authors state that “FMDV genes used in this study were prepared according to a previous report [28],” please describe briefly FMDV DNA templates preparation for the benefits of the readers. Please also consider to test your methodology using DNA viruses as a model in futures. It will help to compare it with a standard PCR.
In order to ensure the reproducibility of the the proposed method it is necessary that the nanconstructs were prepared in a reproducable manner. Please provide the C/Au ratio in obtained AuNPs@GO nanocomposites and comment on the reproducibility of the procedure used to prepare the composite.
The authors devote some space in the article to elucidate mechanism of AuNPs@GO nanocomposite effect on LAMP. The study are based on UV-vis spectroscopy. Unfortunately, these study show only that the individual components of the LAMP cocktail can interact with AuNPs@GO but not much contribute to the explanation of the mechanism itself and molecular base of the observed phenomenon. In particular, the authors do not explain or even hypothesize why the adsorption of LAMP components on the AuNPs@GO surface would have a positive impact on the dynamic processes of interaction of these components taking place during amplification? Please discuss the possible mechanism (s) of the observed enhancement of sensitivity and specificity.
Author Response
Replies to the reviewer 3’s comments
We are thankful for valuable comments of the reviewers, which helped us to correct errors and strengthen up our manuscript.
Reviewer #3 : Authors describe AuNPs@GO nanocomposite assisted LAMP assay for detection of FMDV as a mode to increase sensitivity and specificity of the viral genome detection. The current work is a continuation of the authors' previous studies on using AuNPs@GO as a PCR enhancer. The current work can be a useful contribution to improvement the LAMP, which still needs optimization. I recommend publishing this work with minor corrections.
- The FMDV is a RNA virus. Authors state that “FMDV genes used in this study were prepared according to a previous report [28],” please describe briefly FMDV DNA templates preparation for the benefits of the readers. Please also consider to test your methodology using DNA viruses as a model in futures. It will help to compare it with a standard PCR.
Response : As the reviewer suggested, we added brief description regarding prepation of FMDV genes in the revised manuscript (page 3, line 119-126). Also, FMDV DNA templates were synthesized because we focused on the effects of AuNPs@GO nanocomposite on the step of gene amplification during LAMP reaciton except reverse transcription even though FMDV is an RNA virus. In this regard, our methodology can be expected as a model system for detection of DNA viruses, such as adenovirus in futures.
“ Briefly, sequences of FMDV serotype O and A genes were selected from part of the RNA-dependent RNA polymerase sequences which were originated from O/Andong/KOR/2010 and A/Pocheon/001/KOR/2010 strains, respectively. The serotype O and A genes were cloned into pET-21a plasmids (Novagen, Inc., Madison, WI, USA) and the resulting plasmids were transformed into DH5α E. coli cells (RBC Bioscience Corp., New Taipei City, Taiwan). The transformed cells were grown in a LB medium containing ampicillin (100 μg/ml) at 37 °C overnight. Plasmids were prepared from cultured and harvested cells using extraction kits (Bioneer, Daejeon, Korea).”
- In order to ensure the reproducibility of the the proposed method it is necessary that the nanoconstructs were prepared in a reproducable manner. Please provide the C/Au ratio in obtained AuNPs@GO nanocomposites and comment on the reproducibility of the procedure used to prepare the composite.
Response : The synthesis method of AuNPs@GO nanocomposites used in our experiment has been widely applied in various fields and some references in which this procedure was utilized were already added in the manuscript. Therefore, we believe that the reproducibility of construction manner of AuNPs@GO nanocomposites is certainly guaranteed. In addition, the C/Au ratio of AuNPs@GO nanocomposites is provided as a reference, in which similar protocol was used to synthesize nanocomposites, instead of data of obtained nanocomposites (page 2, line 87-88, reference # 32).
“ The AuNPs@GO nanocomposite with an elemental composition ratio (carbon : gold) of 0.84: 11.7 (wt %) was prepared through in situ growth of AuNPs on GO sheets [30-32].”
- The authors devote some space in the article to elucidate mechanism of AuNPs@GO nanocomposite effect on LAMP. The study are based on UV-vis spectroscopy. Unfortunately, these study show only that the individual components of the LAMP cocktail can interact with AuNPs@GO but not much contribute to the explanation of the mechanism itself and molecular base of the observed phenomenon. In particular, the authors do not explain or even hypothesize why the adsorption of LAMP components on the AuNPs@GO surface would have a positive impact on the dynamic processes of interaction of these components taking place during amplification? Please discuss the possible mechanism (s) of the observed enhancement of sensitivity and specificity.
Response : As the reviewer commented, we added description regarding discussion on the mechanism of enhancement of sensitivity and specificity (page 11, line 303-309).
“ In summary, first, AuNPs@GO can act as a building block, which means that the stability of LAMP components including primers, DNAs and polymerase can be improved through interaction with AuNPs@GO during LAMP reactions. Also, the strong interaction between AuNPs@GO and primers may prevent mismatched hybridization of primers before pairing with target DNAs. Finally, the weaker negative charge of AuNPs@GO may facilitate access of target DNAs or primers than GO, resulting in high sensitivity and specificity. ”
